

# Impacts of the Icelandic Holuhraun volcanic eruption on cloud properties using regional model cloud-aerosol simulations

Masaru Yoshioka[1], Daniel P. Grosvenor[1,2], Amy H. Peace[2,3], Jim M. Haywood[2,3], Ying Chen[3,4] and Paul R. Field[1,2]

[1]School of Earth and Environment, University of Leeds, Leeds, LS2 9JT, United Kingdom
[2]Met Office Hadley Centre, Exeter, EX1 3PB, United Kingdom
[3]Faculty of Environment, Science and Economy, University of Exeter, Exeter, EX4 4QE, United Kingdom
[4]School of Geography, Earth and Environmental Sciences, University of Birmingham, Birmingham, B15 2TT, United Kingdom

*Correspondence to*: Masaru Yoshioka (M.Yoshioka@leeds.ac.uk)

## Abstract

Aerosol-cloud interactions remain a significant uncertainty in climate prediction, largely due to the complexity of measuring and modelling these processes. Volcanic eruptions, such as the Holuhraun event in 2014, offer valuable opportunities to study these interactions by introducing substantial aerosol perturbations. In this study, we investigate the impacts of the Icelandic Holuhraun volcanic eruption on cloud properties using the CASIM cloud microphysics model and the UKCA-GLOMAP aerosol microphysics model within the high-resolution regional model of the UK Met Office Unified Model.

For a four-week simulation, our findings indicate a more than 80% increase in droplet number concentration during the eruption with reductions in cloud droplet sizes, both of which are statistically significant at 0.05 level in t-tests. In contrast, the effects of the volcanic eruption on liquid water path and cloud fraction are not generally significant. During the third week of September, neither satellite observations nor model simulations show significant impacts of the volcanic plume on cloud properties when comparing in-plume to out-of-plume properties. Our simulations suggest that the volcanic aerosol effect during this period was masked by factors affecting the out-of-plume atmospheric conditions, such as natural meteorological variability or non-volcanic aerosols possibly originating from Europe. When volcano on/off simulations are examined, the droplet number increase and the reduction in droplet size remain evident, indicating that these effects are still active. This highlights the crucial role of realistic models in revealing aerosol-cloud interactions that can be obscured in observations due to environmental/meteorological factors.

## 1 Introduction

Aerosol-cloud interactions (ACIs) remain a large uncertainty for climate prediction (Bellouin et al. 2020, Szopa et al. 2021, Watson-Parris et al. 2022). Reasons include i) the difficulty in skilfully predicting the geographical location and vertical profile of the amount, size and composition of aerosols, ii) the complex evolution and feedback of cloud systems when exposed to different aerosol environments, and iii) the challenge of measuring aerosol-cloud effects in the field to test models. To improve climate predictions, aerosol-cloud processes need to be constrained with measurements.

Effusive volcanic eruptions offer a unique opportunity to test and understand biases and uncertainties in aerosol-cloud interaction representations in numerical weather prediction and climate models, as first suggested by Gasso (2008). Volcanic eruptions can introduce large perturbations of aerosol that provide valuable opportunities to study aerosol-cloud interactions. However, while these interactions can be observed from satellite, the complex environment of volcanic plumes and the co-





emission of several different aerosols and their gaseous precursors, makes it difficult to quantify subtle changes. This underscores the need for model simulations to fully understand these processes.

ACIs begin with aerosol acting either as cloud condensation nuclei or ice nucleating particles. In this paper we will only be considering changes to cloud condensation nuclei (CCN) and their interaction with liquid water cloud. The primary aerosol-cloud effect suggested to occur when aerosols are increased is that, for the same mass of liquid water, the droplets will be more numerous but smaller - increasing the albedo of the cloud (Twomey effect, Twomey, 1977). Secondary effects due to changes in droplet sizes can influence precipitation development and cloud evolution. The original Albrecht effect
hypothesis (Albrecht, 1989) was based on the concept that smaller droplets suppress rain formation, as they take longer to evolve into precipitation-sized droplets. Since precipitation acts as a cloud water sink and increases the stability of the boundary layer (Stevens and Feingold, 2009), reduced precipitation through aerosol enhancement can lead to longer cloud lifetimes, increased cloud water contents, and enhanced cloud cover. Consequently, this would also lead to an increase in the albedo of a cloud system. However, these effects of aerosols on cloud properties, known as cloud adjustments, have not been
clearly demonstrated possibly due to local adjustments (Stevens and Feingold 2009) or larger scale circulation modifications (Dagan et al. 2023).

Previously, high-resolution process-based modelling of aerosol-cloud interactions have been explored through the use of box modelling or, if sedimentation is included, 1-d column modelling (e.g. Sorooshian et al. 2010). Some of these modelling studies have formed the basis of bulk activation parametrisations (e.g. Abdul-Razzak and Ghan 2000, Nenes and Seinfeld
2003) in regional and global scale models. However, these approaches are unable to capture feedbacks onto the cloud field from changes in the evolution in the cloud due to aerosol changes. 3-d modelling at hectometre resolution with Large Eddy Models (LES) has sought to explore this (e.g. Jiang et al. 2006). With larger domains it becomes possible to attempt comparisons with observations. Besides volcanoes, shiptracks offer a highly localised anthropogenic aerosol perturbation that has long been observed and studied in satellite images (Gryspeerdt et al., 2021). Comparisons of ship-track observations
with LES were usually idealised and so the impact of the meteorology that the aerosol field is embedded within remained unquantified and uncertain. Larger scale aerosol perturbations from volcanic eruptions have been compared directly with global climate model output (Malavelle et al., 2017, Toll et al. 2017). Using satellite observations of the Icelandic Holuhraun eruption in 2014, Malavelle et al. (2017) determined that, while there was an observed change in the effective radii of cloud particles associated with the plume, the liquid water path and cloud cover did not increase. However, moderate increases in
these quantities would have been impossible to detect amongst their natural variability. The global models did capture the reduction in effective radii, but there was also a wide range in responses for changes in liquid water path related to a potentially erroneous modelling of cloud adjustments.

The Holuhraun eruption has been revisited using regional modelling, using the ICON model in numerical weather forecasting mode at 2.5 km grid resolution (Haghighatnasab et al., 2022). They found when simulating the first week of the
eruption that there was a clear change in droplet number (enhancement of 80%), but no clear signal for liquid water path or cloud fraction due to the compensation between geometrically thick clouds getting thicker and thin clouds getting thinner. Their approach used satellite-derived column SO2 values to scale aerosol reanalysis fields for estimating droplet number concentration rather than explicitly evolving a plume with an aerosol and chemistry scheme.

Further analysis of a 4-week period of the Holuhraun eruption using satellite data (Peace et al., 2024) again indicated that
cloud droplets in the plume decreased in size and became more numerous, apart from one week where there was no obvious in-plume to out-of-plume contrast in effective radius. This was attributed to different airmass histories based on back-trajectory analysis. Comparison to Earth system model (UKESM1: Mulcahy et al., 2020) simulations at N96 resolution (approximately 90 km x 140 km grid spacing around Iceland) showed similar changes in the first week for droplet number concentration and effective radius but less agreement in the following weeks with the model not able to reproduce the
apparent lack of enhancement of cloud droplet number concentrations in the plume.

In contrast to the regional kilometre-scale modelling and satellite analysis, Chen et al. (2022) built a machine-learning model of cloud properties based on meteorological environment. They found an aerosol-induced increase in cloud cover of





approximately 10% by disentangling from meteorological co-variability. This effect was stronger than the impact of increased albedo from the reduction in droplet size that was also predicted by the machine-learning model. The strong cloud
cover increase has also been documented using satellite-based statistical approaches (Wang et al., 2024).

There is a diverse set of responses to the aerosol perturbation from the Holuhraun eruption in global model cloud cover and liquid water path that needs to be constrained to improve the representation of aerosol-cloud interactions in climate models. The analysis of satellite observations by Malavelle et al. (2017) suggests an effective radius response only, whereas the machine learning approach of Chen et al. (2022), that contrasts the volcano-affected observations approach with a surrogate
cloud construction for no-volcano conditions, points to a cloud cover response. Motivated by these differences we make use of the Regional Nested Suite of the UK Met Office Unified Model coupled to the UKCA-GLOMAP aerosol representation (Gordon et al., 2023) to simulate the volcanic plume from the 2014 Holuhraun eruption, its advection and evolution into CCN and the subsequent impact on the cloud fields. We address three key questions related to the Holuhraun volcanic eruption: 1) Was there a change in droplet number and effective radius? 2) Was there a change in cloud cover or liquid water
path? The third question is based on the satellite observations showing a lack of in-plume/out of plume contrast in cloud properties during one week of the eruption: 3) Did any aerosol-cloud effect operate during the period of no apparent plume impact on the clouds?

Section 2 describes the model setup, volcanic plume input to the model, representations of aerosols and cloud, simulations conducted, satellite observations used in this study, and analyses performed. Section 3 describes the main results, and section
4 contains a discussion and conclusions.

## 2 Methods

### 2.1 The host model

In this study we followed the regional Nested UM with Aerosols and Chemistry (NUMAC) model configuration described in Gordon et al. (2023). It is based on the NWP (Numerical Weather Prediction) configuration of the UK Met Office Unified
Model (UM). For the regional atmosphere model, we used the UM version 12.0 nesting suite coupled with submodels to represent processes including atmospheric chemistry, aerosol microphysics as well as cloud microphysics.

A rectangular regional domain of 3500 km x 2750 km centred at 65.0°N, 20.0°W in Iceland has been set to cover the entire northern North Atlantic. The horizontal resolution within the regional domain (or the nest) is 2.5 km. This nesting suite model uses rotated pole coordinates and therefore the sizes of individual grid cells are approximately constant within the
domain. Vertically the model resolves the atmosphere from the surface to 40 km of altitude with 90 layers, 16 layers below 1 km.

The lateral boundary conditions are provided by the Met Office global model (GAL6.1 Walters et al., 2017). The global model has Cartesian coordinates with n216 resolution where grid cells have a size of 0.83 degree in longitude and 0.55 degree in latitude.

### 115 2.2 Models for chemistry, aerosols and clouds

Both global and regional models are coupled with the United Kingdom Chemistry and Aerosols (UKCA) model to simulate atmospheric transport and chemical processes (Abraham, 2014; O'Connor et al., 2014). Within the UKCA model, atmospheric aerosol processes such as new particle formation, gas to particle transfer, coagulation between particles, cloud processing of aerosols, and dry and wet deposition of aerosols are simulated with the modal version of the GLObal Model of
Aerosol Processes model (GLOMAP-mode; Mann et al., 2010).

In the model setup we use in this study, GLOMAP resolves four aerosol components (sulfate, organic carbon, black carbon, and sea salt) in five internally mixed modes (soluble modes in the nucleation, Aitken, accumulation, and coarse size ranges





and an insoluble mode in the Aitken size range). Within each mode a lognormal particle number-size distribution is represented. Emissions and atmospheric processes for mineral dust are treated within the CLASSIC bin scheme (Woodward, 2001) and do not interact with other aerosols and clouds.

The Cloud AeroSol Interacting Microphysics (CASIM; Field et al. 2023) scheme prognoses mass and number mixing ratios of five species of hydrometeor (cloud water, cloud ice, rain, snow and graupel), representing the process rates that move mass and number between species and to and from the vapour phase, as well as handling sedimentation. CASIM uses the Abdul-Razzak and Ghan (2000) bulk activation parametrization to convert UKCA-GLOMAP aerosols into cloud droplets when new water is condensed by the cloud fraction scheme (Gordon et al, 2020). The relationships for autoconversion and accretion, which describe the evolution of cloud water into rain, are based on Khairoutdinov and Kogan (2000). This formulation suppresses autoconversion as droplet number increases.

The cloud fraction scheme (van Weverberg et al., 2021) represents the subgrid humidity distribution and determines when liquid water will condense based on saturation adjustment appropriate for models using timesteps of ~60s. The humidity in the grid box is represented by a Gaussian distribution (two distributions if the grid box is close to an inversion, to represent mixing across the inversion). The width of the distribution is diagnosed from the turbulence closure given in the model for the boundary layer representation (Lock et al., 2000). The scheme can also account for the presence of ice in a grid box that will lead to a narrowing of the subgrid distribution of relative humidity making it more difficult to form supercooled liquid cloud.

### 2.3 Aerosol emissions

Aerosol related emissions including anthropogenic sulfur dioxide ($SO_2$), black carbon (BC), and organic compounds (OC) are taken from CMIP6 data for year 2000 through ancillary files. For the natural $SO_2$ emissions from continually degassing volcanoes, the dataset from Andres and Kasgnoc (1998) is used. The emissions from explosive volcanic eruptions including that from the Holuhraun eruption in 2014 are prescribed separately as described in the next section. Emission of dimethyl sulfide (DMS) from the sea surface is calculated interactively following Nightingale et al. (2000) based on model-predicted surface windspeed and prescribed DMS concentrations in near-surface sea water by Kettle and Andreae (2000). DMS is oxidized to form $SO_2$ which is a precursor of sulfate aerosol (e.g., Fung et al, 2022). Sea salt aerosol is also calculated interactively within the model based on the 10-m windspeed (Gong, 2003).

### 2.4 Holuhraun volcanic emissions

The Holuhraun effusive eruption started on 31st August 2014 and lasted until 27th February 2015, and it injected $SO_2$ into the troposphere at a varying rate. Within our simulations, the strength of the emission has been set to $1.4 \times 10^6$ tonnes of $SO_2$ from 31st August to 13th September 2014 ($10^5$ tonnes per day) followed by $0.98 \times 10^6$ tonnes ($5.4 \times 10^4$ tonnes per day) until 30th of September (Malavelle et al., 2017).

The $SO_2$ is injected into the atmosphere centred at the location of Holuhraun (64.85°N, 16.83°W) and between the altitudes of 800 m and 3000 m. The initial plume is assumed to have a horizontally concentric distribution, with the $SO_2$ concentration highest at the centre and decreasing radially with distance following a Gaussian distribution with a standard deviation of 30 km. $SO_2$ concentration is assumed to be vertically uniform across the injected altitudes.

Following the injection of $SO_2$ into the atmosphere, the plume will evolve as the gas is converted into aerosols, as described in the following sub-section.

### 2.5 Atmospheric aerosol processes

The key processes for this work are summarised here. More comprehensive descriptions of aerosol processes can be found in Yoshioka et al. (2019).



SO2 is oxidized to form sulfuric acid, which is then transferred to the particle phase within the free troposphere through binary homogeneous nucleation. The boundary layer nucleation is turned off in the default version of the model. This process is mediated by secondary organic material (Metzger et al., 2010) that is a product of biogenic volatile organic compounds from land plants (especially forests). Because of this, we have assumed that this process would not be particularly important for our largely oceanic domain and the barren terrain of much of Iceland. Therefore, this process is switched off for the default simulations, but it is included in additional simulations.

Particle growth occurs through sulfuric acid condensing onto existing particles in the atmosphere and through coagulation between particles. Particle growth is represented by an increase of geometric mean diameter and aerosol particles are transferred to a larger aerosol mode once the diameter exceeds the threshold of the mode. When sulfuric acid condenses onto insoluble particles such as BC and OC they become hydrophilic and are transferred to a soluble mode of the appropriate size.

Soluble particles can absorb water from the atmosphere, increasing in size (hygroscopic growth) and can serve as cloud condensation nuclei (CCN) to become cloud droplets when the size, composition (solubility), relative humidity and vertical velocity combine favourably. This is simulated with the Abdul-Razzak and Ghan (2000) bulk activation parametrization.

Aerosol particles are removed from the atmosphere through dry deposition (gravitational settling and turbulent mixing near the surface), nucleation scavenging (activated to form cloud drops and rained out through autoconversion) and impaction scavenging (washed out by rain drops; Slinn, 1982; Mulcahy et al., 2020).

## 2.6 Simulations

The global model was spun-up for 60 days before starting the regional simulations to ensure that the aerosol fields had reached a steady state. Regional simulations were started from 12 UTC of 30th of August 2014 using the output fields from the spun-up global simulation at this time as their initial conditions. The lateral boundary conditions were provided from the global model that was run in parallel with each regional simulation. The meteorology of the global model was initialized at 12 UTC every day from Met Office UM operational analysis and both global and regional simulations were run for 36 hours in each iteration. We discarded the first 12 hours as spin-up and used the simulation output from t=12 to 36 hours (0-24 UTC of the next day) from each iteration. The next iteration starts at 12 UTC using the meteorological fields from operational analysis of the corresponding date and the aerosol fields inherited from the previous iteration.

We ran regional simulations for two scenarios: Volc is the simulation that includes Holuhraun SO2 emission as described in section 2.4 and NoVolc does not include this emission. Simulations were run to 28th September 2014 and we analyzed the simulations from 1st of September.

Since the original simulation underestimated the observed cloud droplet number concentrations as will be shown in section 3, we modified some of the model settings and parameter values to enhance the background aerosols and clouds. Table 1 shows the changes between the default model and the enhanced aerosols and clouds configuration of the model. The model assumes that 2.5% of SO2 is emitted as primary sulphate particles, and this modification changes the modes of these emissions to the smaller Aitken mode from larger size modes in line with Yoshioka et al. (personal communication). However, this does not affect the Holuhraun volcanic emission for which no primary sulphate emission is assumed.



**Table 1. Adjustments to the model to enhance aerosols and clouds**

| Items | Default settings | Enhanced settings |
|---|---|---|
| Variation of updraught velocity (wvarfac) | 1 | 2 (doubled) |
| Boundary layer nucleation | Off | On (Metzger et al, 2010) |
| Primary marine organic aerosol | Off | On |
| Modes of primary sulphate emission from anthropogenic sources | Accumulation and coarse modes | Aitken mode |
| Modes of primary sulphate emission from natural sources | Aitken and accumulation modes | Aitken mode |

**2.7 Satellite observations**

Simulated cloud properties such as cloud droplet number concentration (CDNC), effective radius (Reff), liquid water path (LWP) and cloud fraction (CF) are compared with the satellite observation data from the MODerate resolution Imaging Spectroradiometer (MODIS) onboard the Aqua satellite.

Peace et al. (2024) used the MODIS COSP 1 degree resolution data set (Pincus et al., 2023) to compare with coarse-resolution models. For the higher-resolution simulations presented here, the same retrieved variables are taken from a higher-
resolution satellite dataset, namely the MODIS Aqua Level-2 Collection 6.1 products (Platnick et al., 2015, 2017). Reff, cloud water path, cloud optical thickness and cloud phase are retrieved at 1 km spatial resolution. Cloud fraction is retrieved at 5 km resolution. CDNC is calculating from Reff and cloud optical thickness assuming adiabatic clouds following Quaas et al. (2006) (https://agupubs.onlinelibrary.wiley.com/doi/10.1029/2007JD008962). The Level 2 swath data was aggregated to a 0.5 x 0.5-degree resolution grid for each day. Cloud fraction was obtained from the cloud mask fraction rather than cloud
retrieval fraction (Peace et al. 2024). The cloud properties are examined for marine liquid cloud with cloud top heights between 1 and 5 km.

Satellite observations of cloud properties are subject to uncertainties due to various factors. A comprehensive overview of the uncertainties in the satellite retrieval of CDNC is given in Grosvenor et al. (2018) where the uncertainty of pixel-level CDNC is estimated to be approximately 78%, although the overall uncertainty is likely to be reduced for area-averaged
values. In the MODIS Aqua dataset we used, only data pixels with cloud optical thickness between 4 and 70, and Reff between 4 and 30 µm were retained as these retrievals are most reliable (Quaas et al., 2006).

**2.8 Plume masks**

We masked both simulation outputs and satellite derived data based on column SO2 loading and separated the region into three parts – in-plume, out-of-plume and out-of-bounds.

In the satellite data, we followed the general methodology of Peace et al. (2024) but with the higher resolution data. Column amounts of SO2 were retrieved from the Ozone Mapping and Profiler Suite (OMPS) Nadir Mapper (NM) onboard the NASA-NOAA Suomi National Polar-orbiting partnership (SNPP) satellite (Flynn et al., 2014; Seftor et al., 2014). The region categorized with SO2 loading higher than 1 DU (Dobson Unit) from OMPS was identified as "in-plume". A 3 x 3-




pixel median filter was then applied to minimise the inclusion of isolated grid cells with SO2 > 1 DU that were not likely to
be part of the plume. The median filtering approach has been used to remove random classification errors when detecting
methane plumes (Varon et al., 2018). Next, we added a rectangular boundary enveloping the in-plume region and set the
outside region as "out-of-bounds" and the region within the boundary but out of the plume as "out-of-plume". We ignored
regions over land to avoid potential biases in satellite retrievals and a larger region around the British Isles to avoid
contamination from land sources of SO2.

To create plume masks for the simulations, we regridded the simulated SO2 to a horizontal resolution of ~0.5 degrees,
roughly equivalent to the satellite data. Since using a 1.0 DU criterion of simulated column SO2 resulted in smaller in-plume
regions than satellite derived masks, the threshold was changed the 0.5 DU to make the regions of interest comparable.

In the following sections we will apply satellite derived plume masks to satellite data and simulation derived masks on
simulation data to compare the cloud properties in in-plume and out-of-plume regions. Figure 1 shows the plume masks from
the satellite and the model from 1st to 6th of September 2014 (day 2-7 of the Holuhraun eruption). The masks for the rest of
the simulation period (7th to 28th of September) are shown in figure S1. The masks created from the simulation agree well
with those from the satellite data.

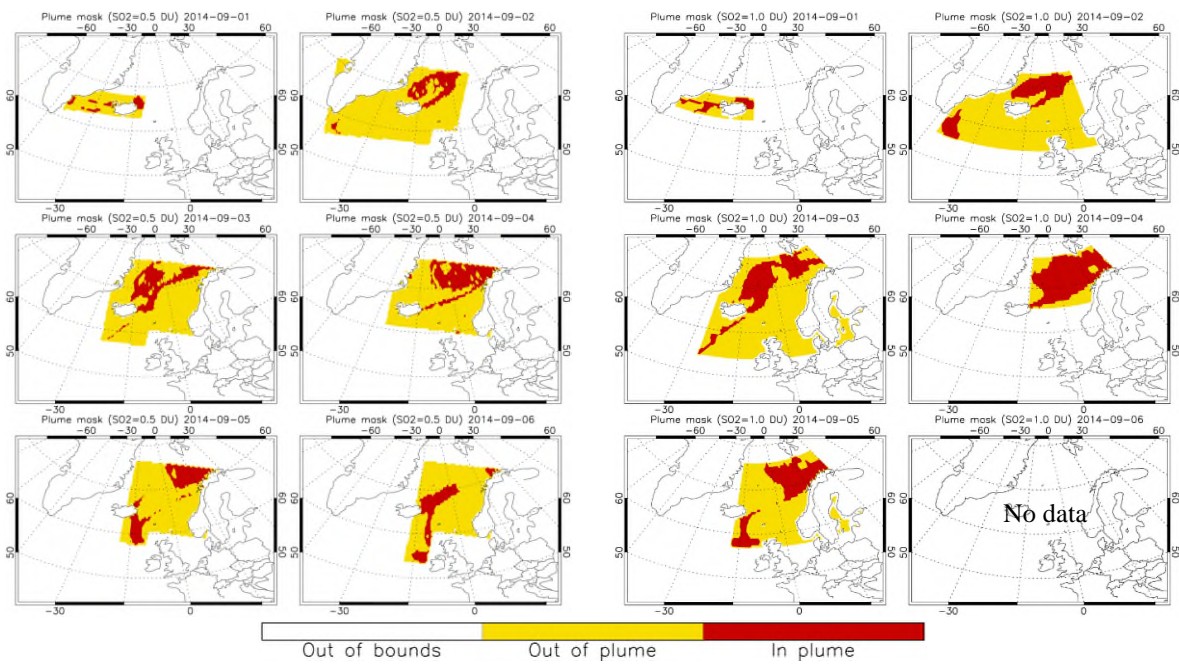


**Figure 1. Plume masks based on the simulated (two columns on the left) and satellite derived (two columns on the right) column
amount of SO2 for 1st to 6th of September 2014 (or day 2 to 7 of eruption). Note that there is no satellite data on 6th of September.
Regions with column SO2 higher than 0.5 DU in the simulation and 1 DU with a median filter in the satellite data are classified as
"in plume". See the text in Section 2.9 for more info. Plume masks for 7th to 28th of September are shown in figure S1.**

**2.9 Post processing of data**

Column mean CDNCs in the simulations were calculated by vertically averaging CDNCs within grid boxes using liquid
water contents as weights to account for the vertically varying amounts of cloud condensate. This approach will weigh the
value towards cloud top to be more comparable with the satellite view of the cloud. Cloudy-sky LWPs were obtained by
dividing column LWPs by column cloud fraction. Column volume mean cloud droplet radii were calculated by dividing





column LWPs by total column CDNCs, dividing by the density of water and converting from the resulting volume to a radius assuming spherical droplets. The volume mean radius was converted to Reff by multiplying by 1.24. This follows from CASIM assuming the cloud droplet distribution follows a gamma distribution with a shape factor of 2.5. The ratio of effective radius (ratio of third and second moments of the droplet distribution) to the volume mean radius (ratio of third to zeroth moment all to the third power) is then 1.24.

To carry out comparisons with satellite observations (MODIS AQUA), the horizontal resolutions of simulated cloud properties were reduced to about 50 km by horizontally averaging the values within the box of reduced resolution. To account for the insensitivity of the satellite instrument to thin clouds, all data where column LWP is less than 10 g m-2 were excluded.

The data were split into four one-week periods for comparing to the satellite data following Peace et al. (2024).

**2.10 Effects of volcanic emissions on clouds**

We split observed and simulated data into in-plume and out-of-plume using the plume masks as described above. For the observations and the Volc simulation we call the difference between a cloud property in and out of the plumes as the 'TOTAL effect'. We calculate this either as the difference between two arithmetic means or the quotient between two geometric means depending on the distribution of the variable. This is the effect that can be observed with satellites.

In simulations we can use the counterfactual NoVolc scenario to estimate the effects of different background meteorology and other environmental factors depending on the location of the plume. We call this the 'LOCATION effect' and calculate it as the difference between arithmetic means, or the quotient between geometric means, inside and outside the plumes in the NoVolc simulation.

The difference between total effect and location effect is considered to be caused by the volcanic plume. In our model we
only take into account sulphate aerosol produced by SO2 as the impact of the volcanic emission. We call this the 'AEROSOL effect' and calculate it as the difference or the quotient, where appropriate, between the TOTAL effect and the LOCATION effect. We also look at the difference between the Volc in-plume and NoVolc in-plume to obtain the 'ERUPTION effect'. This allows us to see if there is an impact of the aerosol on the cloud fields and avoids any impact of the meteorology acting to mask any effects in the satellite data. The AEROSOL effect converges to the ERUPTION effect if
the region outside the plume remains unaffected by the volcanic eruption (i.e., there is no 'REMOTE effect'). Reasons for these values being different include diffuse plumes below the given threshold, changes in circulation caused by modifications to heating rate profiles in the plume region that impact cloud fields outside of the plume, or the presence of large amounts of additional water vapour in the plume that is not included here. TOTAL, LOCATION, ERUPTION, REMOTE, and AEROSOL effects are summarised in figure 2 and table 2.



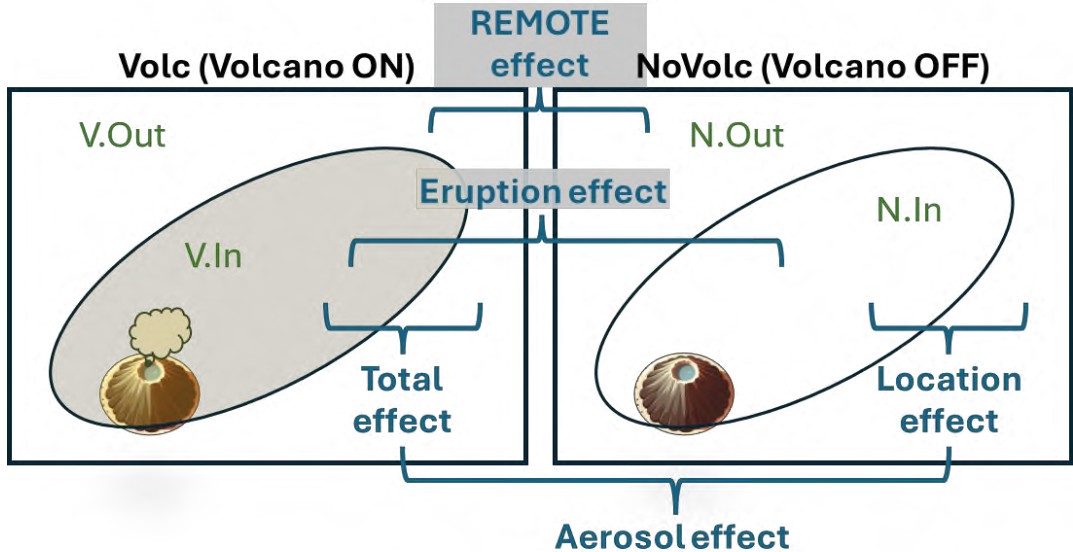


**Figure 2. Schematic diagram of TOTAL, LOCATION, ERUPTION, REMOTE, and AEROSOL effects.**

**Table 2. Formulations for TOTAL, LOCATION, ERUPTION, REMOTE, and AEROSOL effects. 'O', 'V', and 'N' represent the observations and Volc and NoVolc simulations, respectively. 'In' and 'Out' indicate inside and outside of plume, respectively. '.vs.'**
**means either '–' (minus) for Reff and CF where linear statistics are used or '/' (divided by) for CDNC and LWP where geometric statistics are used.**

| Effect | Formula | Observable? |
|---|---|---|
| **TOTAL effect** | = O.In *vs.* O.Out for observation<br>= V.In *vs.* V.Out for simulation | Yes |
| **LOCATION effect** | = N.In *vs.* N.Out | No |
| **ERUPTION effect** | = V.In *vs.* N.In | No |
| **REMOTE effect** | = V.Out *vs.* N.Out | No |
| **AEROSOL effect** | = [TOTAL effect] *vs.* [LOCATION effect]<br>= [V.In *vs.* V.Out] *vs.* [N.In *vs.* N.Out]<br>= [V.In *vs.* N.In] *vs.* [V.Out *vs.* N.Out]<br>= [ERUPTION effect] *vs.* [REMOTE effect] | No |



# 3 Results

## 3.1 Base model results

We will discuss the results in the following order. First, the aerosol is perturbed by the volcanic plume and is expected to feed through directly to CDNC. Secondly, changes in CDNC control the process that converts cloud water to rain affecting the LWP, which is therefore examined next. Thirdly, Reff, that impacts the radiation, is diagnosed from the primary characteristics predicted by the model (liquid water mass and droplet number concentration). Finally, the emergent behaviour of the cloud cover that results from all of the changes is examined.

Figure 3 shows the values of observed and simulated CDNCs at three quartiles (25th, 50th and 75th percentiles) as well as the 5th and 95th percentiles within and out of the volcanic plumes during the four weeks from the start of the eruption. The probability distribution functions for the same data are shown in figure S2. CDNC is underestimated in the simulation including volcanic emissions (Volc; red) both within (dark colours) and out of (light colours) the plume by factors of 2-4 compared to the observation (grey). Enhancement of CDNC within the plume compared to outside of the plume can be seen

in both the observations and in the Volc simulation in all but the third week. Since CDNC is underestimated outside the plume and the magnitudes of enhancements within the plumes are about right, the general underestimation is considered likely due to biases in background aerosols and clouds rather than a bias in the Holuhraun volcanic emission implemented in the model.





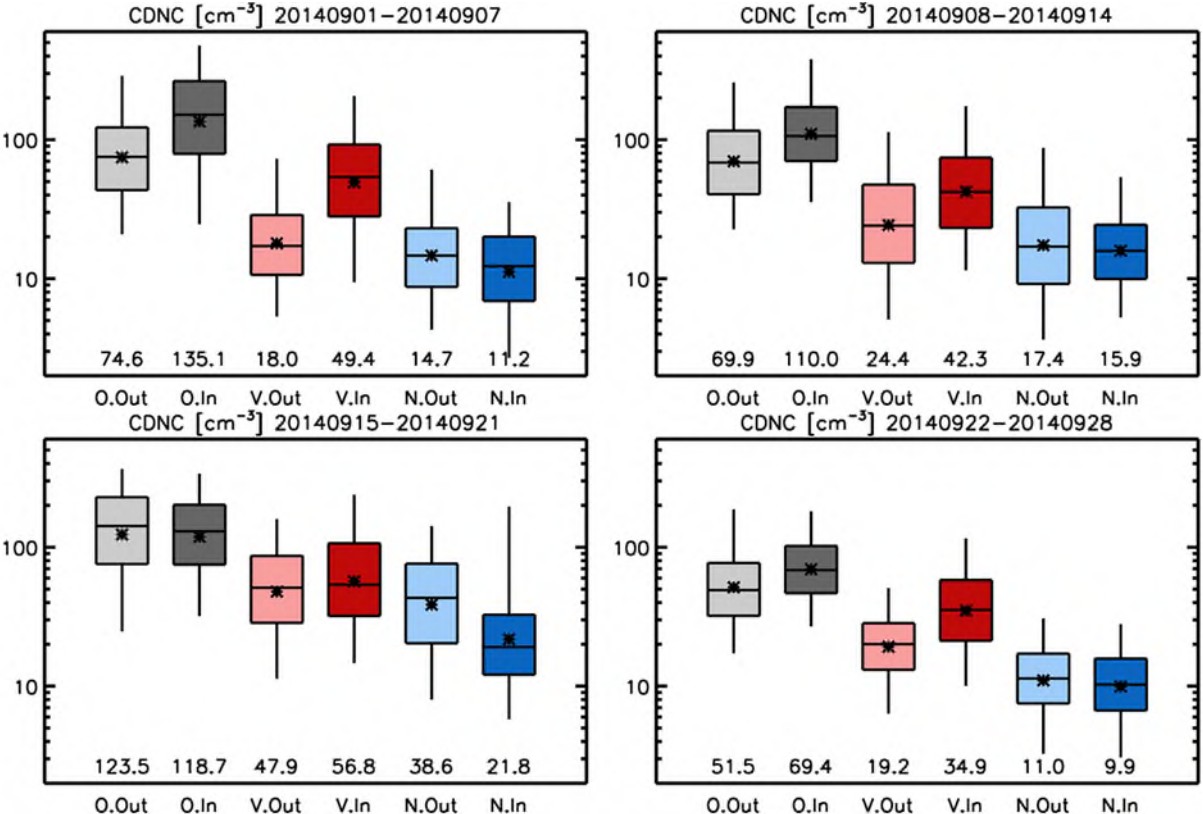

**Figure 3. Box and whisker plots showing the values of CDNC at three quartiles (25th, 50th and 75th percentiles) and the 5th and 95th percentiles for the four weeks from the start of the eruption. Results are shown for the observations ('O' in x-axis and grey), the Volc simulation ('V' and red) and the NoVolc simulation ('N' and blue) within ('In' in x-axis and dark colours) and out of ('Out' and light colours) the volcanic plume judged by column SO2 loading. The geometric means are shown as stars and values near the bottom of the frames. Note that the vertical axis has a log scale. Corresponding probability distribution functions are shown in figure S2.**



## 3.2 Results from the simulations with enhanced aerosols

Following the relatively large underestimates in simulated background CDNC, we modified the model settings as described
in section 2.6 and all the subsequent results use this new configuration. Figure 4 replicates figure 3 but for the simulations
with enhanced settings (Volc: V and NoVolc: N). The background values of CDNC have been improved and the
underestimates in the Volc simulation are within a factor of ~2 and in most cases less than 1.5 (50%).

CDNC is enhanced within the plumes in both the observations and the Volc simulation except for during the third week. As
shown in table 3 and figure 8, student t-tests indicate a statistically insignificant (at 0.01 level) difference between the CDNC
within and out of plume (TOTAL effect) for the third week for both the model and observations. Note that t-tests were
performed on the base 10 logarithms of the values and a lagged autocorrelation has been performed to estimate the degrees
of freedom by assessing the scale at which the field becomes decorrelated (e.g. Field and Wood 2007). The ERUPTION
effect for the model (figure 4, table 3, figure 8) shows that the aerosol perturbation in the plume region always leads to an
enhancement of CDNC (factors of 2.4-4.2). The LOCATION effects are significant for weeks 1 and 3 but no significant
differences for weeks 2 and 4. In contrast, the REMOTE effects are significant in all weeks.

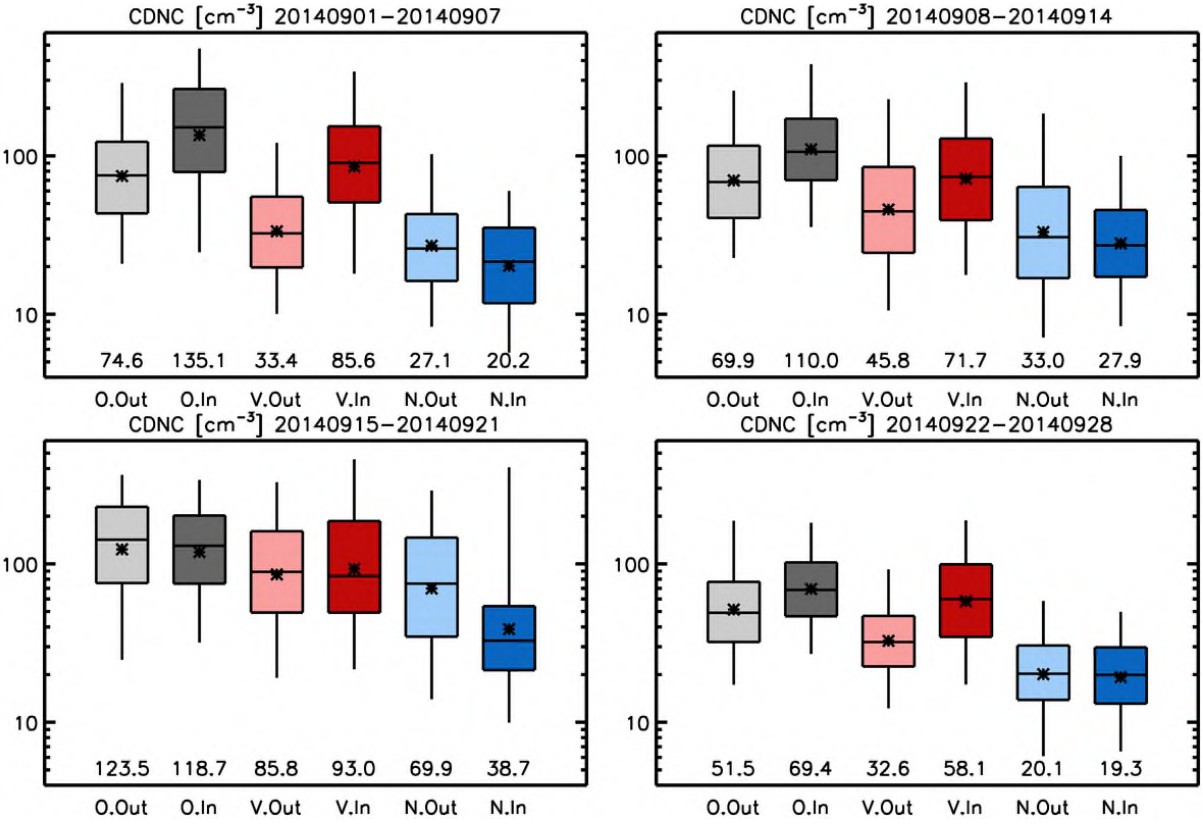


**Figure 4. The same as figure 3 but for simulations with enhanced aerosols and clouds. Note that the vertical axis has a log scale but
has been changed from figure 3. Values at the bottom of each panel show geometric means. Corresponding probability
distribution functions are shown in figure S3.**

Cloudy-sky LWP (figure 5) is also underestimated by factors up to ~2 compared to the satellite observations, although
MODIS LWP can be in error due to various factors, such as effective radius and optical depth retrieval errors, and





assumptions regarding the degree of adiabaticity used to estimate LWP. Notwithstanding systematic errors in the MODIS LWP retrieval, table 3 shows that observed differences between within- and out-of-plume (TOTAL effect) are significant in the first 3 weeks with increased LWP in the plume for weeks 1 and 2 but decreased LWP in the plume for week 3. In the model, on the other hand, the TOTAL effect is not deemed significant for any of the weeks. The ERUPTION effect (V.in versus N.in) is about +20% and statistically significant in the first week at the 0.05 (but not 0.01) level, while the other weeks indicate changes of ~5% that are not deemed significant. The LOCATION effect is significant in the first week at 0.01 level (significant at 0.05 level in the third week), while the REMOTE effect does not show any significance.

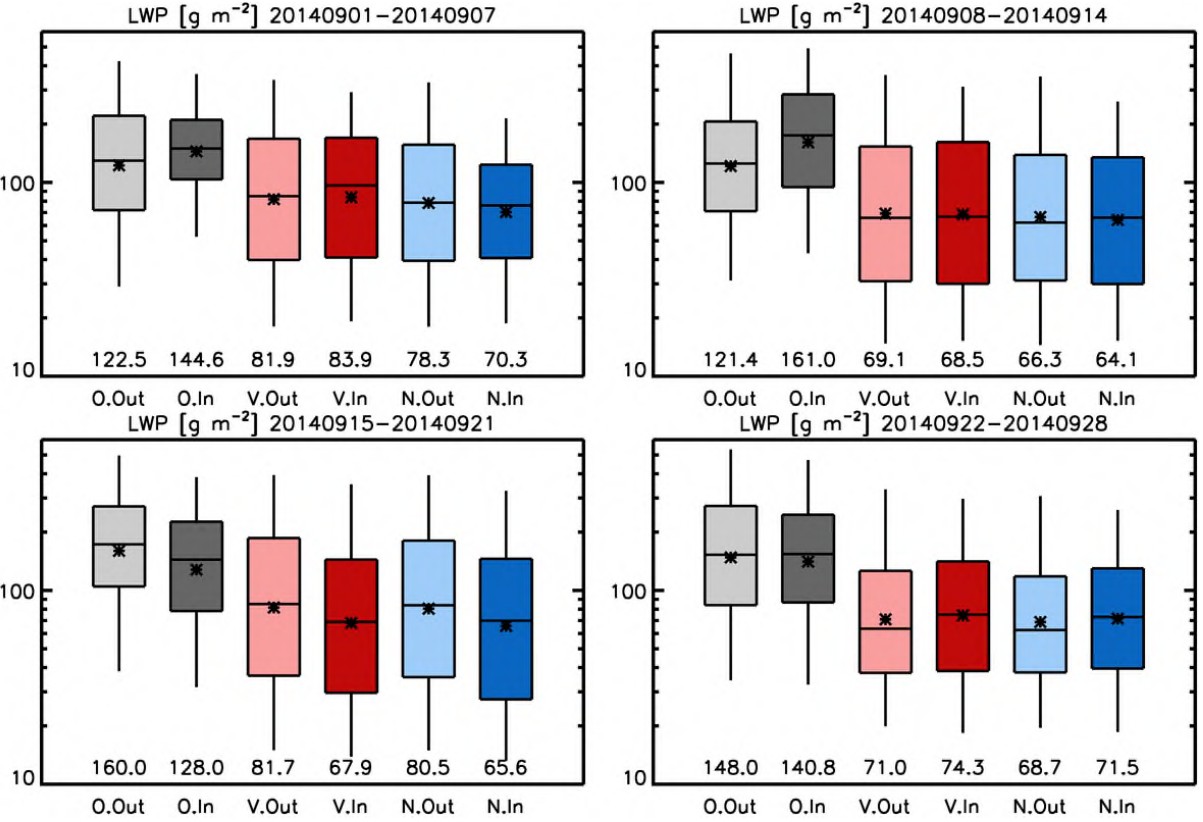

**Figure 5. Same as figure 4, but for liquid water path in the cloudy sky. Corresponding probability distribution functions are shown in figure S4.**




Background Reff (figure 6) are underestimated by up to about 30% (O.Out vs. V.Out). But the differences between within
and out-of-plume values (TOTAL effect) for models and observations are similar, including the weak response for the 3rd
week. This mirrors the CDNC response since, for a constant LWP, Reff will scale inversely with CDNC1/3 and within and
out-of-plume LWP values are very similar in the model. For the observations there are LWP increases in weeks 1 and 2, but
there is still a reduction in Reff for these weeks indicating that the increase in CDNC dominates over the LWP increase in
terms of causing the Reff response. Following the change in CDNC, Reff is also significantly reduced for the eruption effect
(factors of 0.65-0.77). As was the case for the CDNC, Reff shows a non-zero remote effect of up to -15% that is deemed
significant. The location effect again follows the CDNC response with significant differences in the 1st and 3rd weeks.

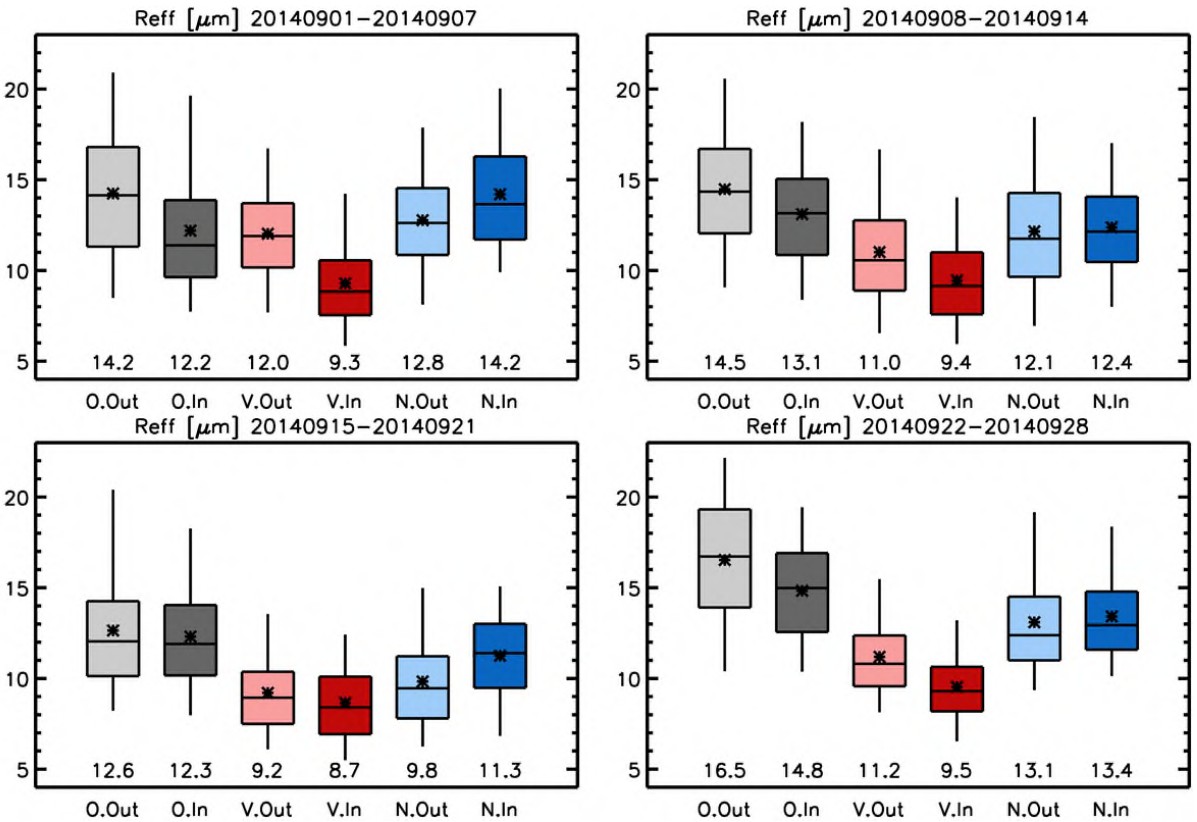

**Figure 6. Same as figure 4, but for droplet effective radius. Note that the vertical axis has a linear scale and the values at the
bottom and shown in stars are arithmetic means. Corresponding probability distribution functions are shown in figure S5.**





Figure 7 indicates that there is 3-5% increase in observed CF within plume region compared to out-of-plume region, except in the 3rd week. On the other hand, the model shows smaller changes in CF. Even when only considering the liquid cloud fraction (see figure S3) there is no significant change in the cloud fraction due to the volcano with the exception of the 3rd week in direct contrast to the observations. The ERUPTION, LOCATION and REMOTE effects all exhibit no significant change.

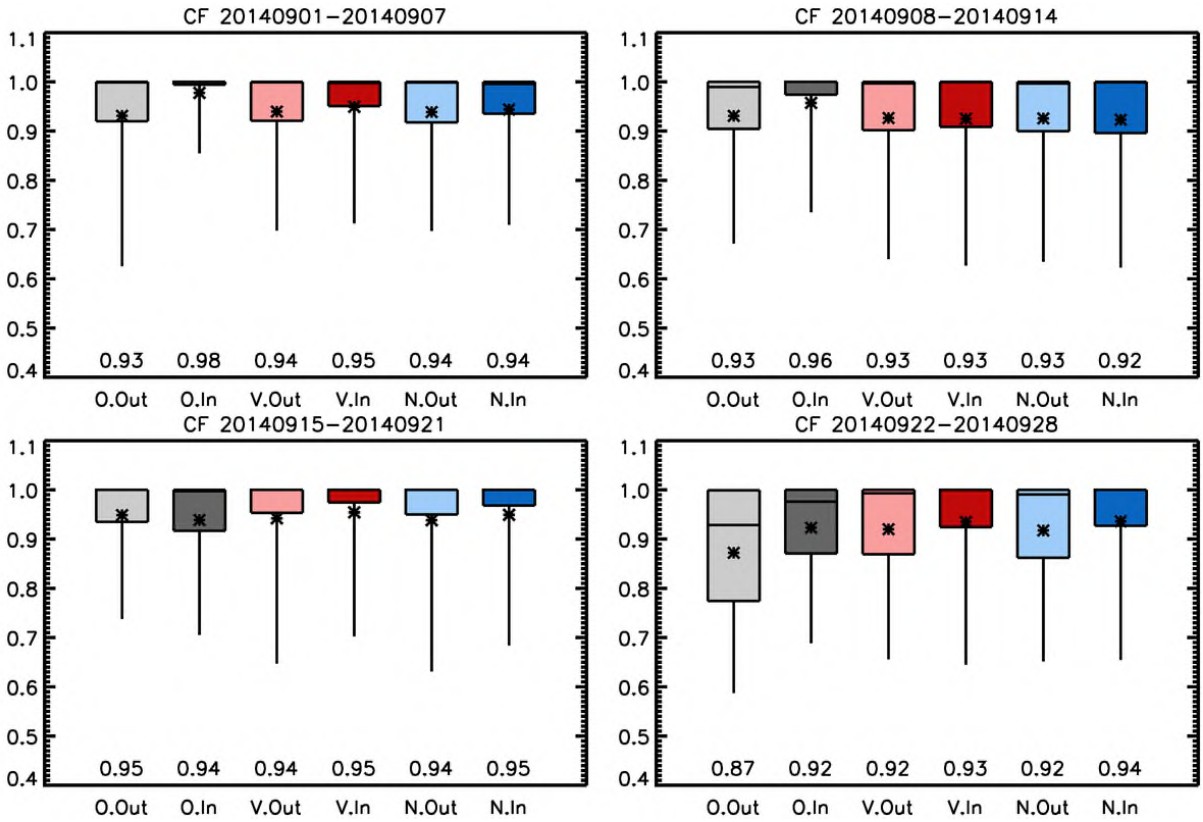


**Figure 7. Same as figure 4, but for cloud fraction. Note that the vertical axis has a linear scale and the values at the bottom and shown in stars are arithmetic means. Corresponding probability distribution functions are shown in figure S6.**





**Table 3. TOTAL, ERUPTION, LOCATION, REMOTE, and AEROSOL effects and corresponding p-values from t-tests (in parentheses) for CDNCs, LWPs, Reffs, and CFs. Shaded values indicate statistically significant differences between the means of two given distributions at the 0.01 level, while p-values exceeding 0.01 and 0.05 are marked with single and double underlines, indicating statistical insignificance at each level. Note that for CDNC and LWP, effects and t-tests are based on geometric means and the base 10 logarithms, respectively. For Reff and CF, they are based on arithmetic means and their values themselves. Model and satellite fields contain significant correlation between neighbouring grid points. Therefore, to obtain a more realistic estimate of the degrees of freedom used to compute the t-statistic the model field have been horizontally lagged until autocorrelation was reduced to 0.3 (e.g. Field and Wood 2007). This lagging (~150-200km) was used to combine independent samples to calculate the t-statistic.**

| | | Observations | Simulations | | | | |
|---|---|---|---|---|---|---|---|
| | Week | TOTAL effect O.In vs. O .Out | TOTAL effect V.In vs. V.Out | ERUPTION effect V.In vs. N.In | LOCATION effect N.In vs. N.Out | REMOTE effect V.Out vs. N.Out | AEROSOL effect TOTAL vs. LOCATION effects |
| CDNC [factor] | 1 | 1.81 (0.000) | 2.56 (0.000) | 4.24 (0.000) | 0.75 (0.000) | 1.23 (0.000) | 3.44 (0.000) |
| | 2 | 1.57 (0.000) | 1.57 (0.000) | 2.57 (0.000) | 0.85 (0.025) | 1.39 (0.000) | 1.85 (0.000) |
| | 3 | 0.96 (0.526) | 1.08 (0.934) | 2.40 (0.000) | 0.55 (0.000) | 1.23 (0.000) | 1.96 (0.000) |
| | 4 | 1.35 (0.000) | 1.78 (0.000) | 3.01 (0.000) | 0.96 (0.019) | 1.62 (0.000) | 1.86 (0.000) |
| LWP [factor] | 1 | 1.18 (0.000) | 1.02 (0.149) | 1.19 (0.011) | 0.90 (0.002) | 1.05 (0.055) | 1.14 (0.013) |
| | 2 | 1.33 (0.000) | 0.99 (0.497) | 1.07 (0.375) | 0.97 (0.283) | 1.04 (0.446) | 1.03 (0.540) |
| | 3 | 0.80 (0.001) | 0.83 (0.053) | 1.04 (0.759) | 0.81 (0.044) | 1.01 (0.726) | 1.02 (0.826) |
| | 4 | 0.95 (0.477) | 1.05 (0.724) | 1.04 (0.602) | 1.04 (0.785) | 1.03 (0.367) | 1.01 (0.145) |
| Reff [μm] | 1 | -2.05 (0.000) | -2.73 (0.000) | -4.91 (0.000) | 1.43 (0.001) | -0.75 (0.000) | -4.16 (0.000) |
| | 2 | -1.37 (0.000) | -1.56 (0.000) | -2.92 (0.000) | 0.22 (0.639) | -1.14 (0.000) | -1.78 (0.000) |
| | 3 | -0.33 (0.203) | -0.55 (0.042) | -2.60 (0.000) | 1.42 (0.000) | -0.63 (0.000) | -1.97 (0.000) |
| | 4 | -1.70 (0.000) | -1.66 (0.000) | -3.88 (0.000) | 0.32 (0.019) | -1.90 (0.000) | -1.98 (0.000) |
| CF [fraction] | 1 | 0.047 (0.000) | 0.010 (0.842) | 0.005 (0.262) | 0.005 (0.422) | 0.000 (0.680) | 0.004 (0.575) |
| | 2 | 0.026 (0.003) | -0.002 (0.509) | 0.002 (0.884) | -0.003 (0.603) | 0.001 (0.981) | 0.001 (0.693) |
| | 3 | -0.010 (0.254) | 0.012 (0.002) | 0.005 (0.717) | 0.011 (0.011) | 0.004 (0.813) | 0.001 (0.930) |
| | 4 | 0.051 (0.000) | 0.015 (0.266) | -0.002 (0.819) | 0.019 (0.080) | 0.002 (0.669) | -0.004 (0.305) |



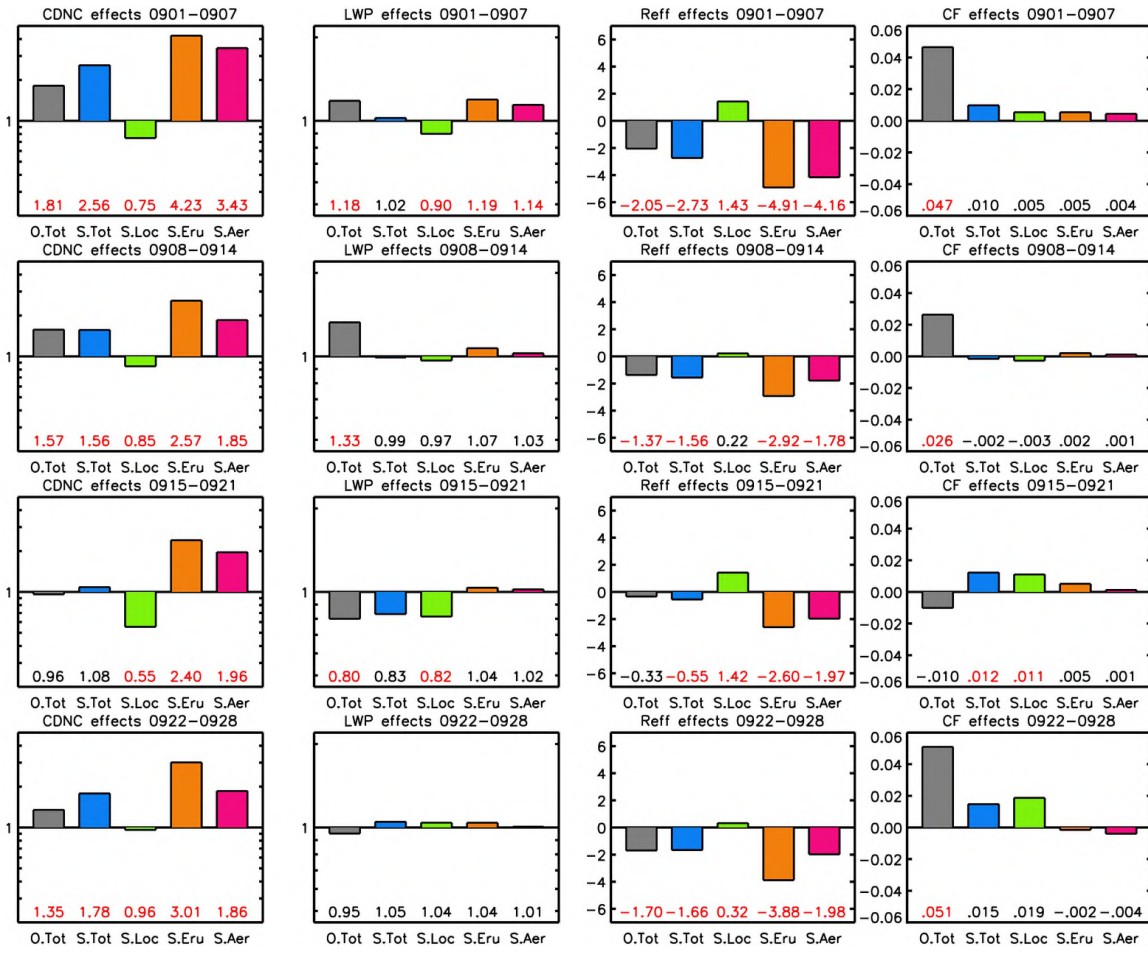

**Figure 8. Mean TOTAL effect of the volcanic plume for the observations (O.Tot) and the simulations (S.Tot), and mean LOCATION, ERUPTION, and AEROSOL effects for the simulations (S.Loc, S.Eru, and S.Aer). See text for descriptions. Results are shown for CDNC, LWP, Reff, and CF (columns) and for the first four weeks after the eruption (rows). Note that base 10 log y-axis scales and enhancement ratios (e.g., TOTAL effect for the simulation = V.In/V.out, see table 2) are used for the CDNC and LWP effects, whereas linear scales and subtracted values (e.g., V.In minus V.out) are used for Reff and CF. Means are geometric for CDNC and LWP and arithmetic for Reff and CF. Values at the bottom of the panels show the mean effect values (same as the bars) and are red if the mean effect is significant at 0.05 level in t-test (see table 3).**



## 4 Discussion and Conclusions

The Holuhraun volcanic eruption in 2014 provided a unique opportunity to explore and gauge the impact of a large aerosol
perturbation on cloud systems in the North Atlantic basin. This long-lasting event has been well documented with satellite
observations and previously modelled with global and regional simulations. Here we used the UK Met Office Unified Model
in regional nesting configuration (2.5 km grid resolution) coupled with the UKCA chemistry and aerosol scheme and a
simple gaussian plume injection of SO2 to evolve the plume and look at the impact of the aerosol in a realistic environment.
This is perhaps the first time a simulation has looked at aerosol-cloud interactions from gaseous volcanic emissions through
aerosol production to cloud droplet activation and cloud system evolution in a realistic regional convection permitting
numerical weather prediction setting.

Observations are necessarily limited by only being able to compare within to out-of-plume values under the assumption that
the meteorological conditions inside and outside the plume are the same and so any differences are solely attributed to the
aerosol perturbation. In contrast, model experiments allow us to take into account the differences in meteorological
conditions with and without volcanic plumes and between the within- and out-of-plume regions to isolate the effect of
volcanic aerosols. Previous machine learning approaches (Chen et al. 2022) attempted to avoid the in-plume/out-of-plume
assumption by building a multi-dimensional response of clouds to the meteorology that can predict the cloud characteristic in
the absence of the volcano. Nevertheless, there is potential for error if there are systematic biases present in the training
dataset. The comparison between satellite observation and the machine-learned surrogate satellite has minimize the impacts
of systematic bias (Chen et al., 2022), future machine-learning with multiple satellites observations (when they are ready)
will further improve the robustness of outcomes.

Table 4 shows results from the MODIS observational data used in this study (see Section 2.8) alongside that from Peace et
al. (2024) in which coarser resolution (1x1 degree) MODIS data was used. Comparing the MODIS results from this study to
those of Peace et al. (2024) shows that they are broadly similar with CDNC enhancements agreeing to within 0.2, LWP
matching closely, Reff matching within 0.05 and CF within 0.03. This provides a measure of observational uncertainty
introduced by using different MODIS aggregated datasets and analysis. Model results from a global model (UKESM1,
1.9x1.3 degree grid resolution) and regional model (ICON, 2.5 km grid resolution) are also shown for comparison to the high
resolution UM used in this study, where available.

The observed CDNC is enhanced within the plumes in all but the third week and this is captured and explained by the
simulation. TOTAL effects for weeks 1, 2 and 4 are between 1.35 and 1.81 in observations and between 1.56 and 2.56 in the
simulations. In contrast, there is almost no enhancement in the third week as shown in the total effects of near unity in both
observation (0.96) and simulation (1.08). The LOCATION effect for CDNC is 0.55 (table 3; figure 8) in the third week and
this indicates a reduction of about a factor of 2 within the plume because of the meteorological and environmental factors
other than the aerosol plume. The model simulations indicates that the AEROSOL effect on CDNC enhancement is 1.96 (the
ERUPTION effect is 2.40). This suggests the enhancement of CDNC by a factor of 2 within the plume was almost
completely offset by the LOCATION effect to yield near unity TOTAL effect in this period. The UKESM TOTAL effect is
correlated with the UM but is smaller in magnitude and, in contrast to the high-resolution UM, shows a reduction in weeks 3
and 4. For the first week, ICON shows a TOTAL effect between that of the UKESM and the high-resolution UM.
Comparing the AEROSOL effect shows that, as would be expected from the large aerosol input from the volcano, the UM is
always enhanced by a large factor (>1.85) while the ICON and UKESM have smaller enhancement factors with close to no
aerosol effect in week 3 (1.05) for the UKESM. The magnitude of the modelled enhancements for CDNC in this study are
larger than the observations. Modifying the base model configuration to allow more background aerosol not only led to an
improved comparison to the observed background droplet concentrations but also reduced the magnitude of the enhancement
impact of the aerosol perturbation (TOTAL effects obtained from the base model simulations are 2.74, 1.73, 1.19, and 1.82
for the four weeks compared to 2.56, 1.57, 1.08 and 1.78 for the enhanced aerosol configuration). Future work on improving
the state of the background aerosol environment and processes may lead to a further reduction in enhancement factor for
CDNC that would agree even more closely with the observations.



For LWP, while the MODIS results indicate a TOTAL enhancement in week 1 and 2, there is a reduction in week 3 and a close to neutral response in week 4 (0.95). The UM and UKESM show similar total effects for weeks 1 and 2, but in week 3
the UKESM shows a stronger reduction (0.69) than the high-resolution UM (0.83) for which none of the changes were deemed significant. The aerosol effect shows that both the UM and UKESM show a similar muted response with the exception of week 1 where slight enhancements (≥1.1) are shown by the UM and UKESM with ICON indicating a larger response (1.23). Reasons for not seeing a LWP effect are potentially linked to an underestimate of LWP by a factor of up to 2 which may be reducing or stalling precipitation formation in the modelled clouds. If precipitation is already suppressed
then increased CCN availability will not have a large effect on precipitation and little or no increase in LWP via the Albrecht effect will result.

Overall, the high-resolution UM shows a much greater TOTAL response (reduction) in Reff than the UKESM, and both are larger than the ICON response in week 1. Similar to the result for CDNC, the UKESM shows little AEROSOL effect in week 3 for Reff, in contrast to the UM, which shows an AEROSOL effect as strong as that for weeks 2 and 4.

For CF the observational datasets hover around 1.0 with variations between the different modelling results of 0.02 to 0.07 with the Peace et al. (2024) coarser data showing some reductions in cloud cover at the same time as the higher resolution data analysis shows slight increases highlighting the difficulty in assessing aerosol impacts on this variable. The model enhancements are not deemed significant. We note that Koren et al. (2007) demonstrated that CF estimates based on AOD thresholds were difficult due to the presence of humidified aerosol close to clouds. Any apparent change in CF associated
with changes in aerosol may reflect this potential misidentification of cloud. Furthermore, Mieslinger et al. (2022) noted that greater aerosol loading could lead to increased cloud reflectivity that increases the probability of previously undetected optically thin clouds becoming optically thick enough to be detected as cloud in CF satellite products. Chen et al. (2022) found with their machine learning model that increasing aerosol could lead to increased CF, but it is unclear if the effects highlighted by Koren et al. (2007) and Mieslinger et al. (2022) could be contributing to this result, and further study with
multiple satellites to test cloud-resolving physical models and explore the underlying processes is warranted. The UKESM total effect shows large changes from enhancements of 1.85 in week 1 to reductions of 0.55 in week 3. In stark contrast the UM does not show much change, although it should be noted that cloud cover is close to maximum for the UM. However, the aerosol effect changes are more similar.



**Table 4. Simulated in-plume enhancements (TOTAL effects) in the first 4 weeks of September 2014 in this study (UM) compared with the observed enhancements from the 1 degree MODIS data presented in Peace et al. (2024; P24), those from the 0.5 degree MODIS data produced for this study (Y25), as well as UKESM1 results (grid resolution 1.9x1.3 degrees). Both the TOTAL effect (ratio of values between inside and outside plume) and the AEROSOL effect values are shown. For P24, the AEROSOL effect was calculated using the values in their table 1 as (in-plume enhancement in 'Hol' (%)/100 + 1)/(in-plume enhancement in 'Ctrl' (%)/100 + 1), where 'Hol' is their simulation with the Holuhraun volcano on and 'Ctrl' is their simulation with no volcano. ICON simulation (grid resolution 2.5km) results for the first week are also included from Haghighatnasab et al. (2022). Note that the TOTAL and AEROSOL effects for all variables are presented as ratios here, including those for Reff and CF, which were shown as differences in Section 3, in order to facilitate comparisons with P24 and ICON.**

| Variable | Effect | Dataset and Study | Week 1 | Week 2 | Week 3 | Week 4 |
|---|---|---|---|---|---|---|
| CDNC | TOTAL | MODIS Y25/P24 | 1.81 / 1.58 | 1.57 / 1.56 | 0.96 / 0.84 | 1.35 / 1.32 |
| | | UM | 2.56 | 1.56 | 1.08 | 1.78 |
| | | UKESM P24 | 1.56 | 1.16 | 0.8 | 0.92 |
| | | ICON | 1.77 | —— | —— | —— |
| | AEROSOL | UM | 3.43 | 1.85 | 1.96 | 1.86 |
| | | UKESM P24 | 1.95 | 1.18 | 1.05 | 1.24 |
| | | ICON | 1.77 | —— | —— | —— |
| LWP | TOTAL | MODIS Y25/P24 | 1.18 / 1.11 | 1.33 / 1.20 | 0.80 / 0.87 | 0.95 / 0.89 |
| | | UM | 1.02 | 0.99 | 0.83 | 1.05 |
| | | UKESM P24 | 1.03 | 0.95 | 0.69 | 0.92 |
| | | ICON | 1.30 | —— | —— | —— |
| | AEROSOL | UM | 1.14 | 1.03 | 1.02 | 1.01 |
| | | UKESM P24 | 1.10 | 1.04 | 0.99 | 1.08 |
| | | ICON | 1.23 | —— | —— | —— |
| Reff | TOTAL | MODIS Y25/P24 | 0.86 / 0.91 | 0.90 / 0.91 | 0.98 / 1.02 | 0.90 / 0.86 |
| | | UM | 0.78 | 1.11 | 0.96 | 0.85 |
| | | UKESM P24 | 0.86 | 0.95 | 1.06 | 0.97 |
| | | ICON | 0.93 | —— | —— | —— |
| | AEROSOL | UM | 0.65 | 0.76 | 0.77 | 0.71 |
| | | UKESM P24 | 0.83 | 0.95 | 0.98 | 0.94 |
| | | ICON | 0.93 | —— | —— | —— |
| CF | TOTAL | MODIS Y25/P24 | 1.05 / 1.03 | 1.03 / 0.96 | 0.99 / 0.97 | 1.05 / 1.02 |
| | | UM | 1.01 | 0.99 | 1.01 | 1.01 |
| | | UKESM P24 | 1.85 | 0.79 | 0.55 | 0.8 |
| | | ICON | 1.40 | —— | —— | —— |
| | AEROSOL | UM | 1.00 | 1.00 | 1.00 | 1.00 |
| | | UKESM P24 | 1.07 | 1.07 | 0.98 | 0.91 |
| | | ICON | 1.06 | —— | —— | —— |

It is of interest to note that N.Out and V.Out values (REMOTE effect) are not identical. The most likely cause of this difference is the presence of volcanic SO2 in low concentrations outside of the plume mask in model simulations, but





changes occurring within the plume region may also contribute by affecting the cloud field further away from the aerosol plume. This could be related to changes in heating profiles caused by cloud albedo changes modifying atmospheric circulations and subsequent cloud evolution close to the vicinity of the plume. The grid resolution of the model may alias
those effects on to larger scales than in reality. To test that hypothesis the study would have to be carried out with much increased resolution that we will leave for later work.

Returning to answer the three questions raised in the introduction:

1. The modelling results show an increase in droplet number and reduction in effective radius, in agreement with observations and previous modelling.

2. The UM liquid water path remains largely unchanged when comparing out of plume to in-plume values but does show a ~10% enhancement for volcano on versus volcano off in the first week in the plume region. The total cloud cover, which is close to totally overcast, is not significantly (from t-test) affected by the volcano. This is different to the observations and may result from lower LWP being predicted in the model for the background state or potential problems with retrieval of CF from satellite.

3. The model reproduced the observations in that, during the third week, in-plume values vs. out-of-plume values for CDNC and Reff values showed little contrast in comparison to weeks 1,2 and 4. The high-resolution modelling indicated that, during week 3, differences in the meteorological environment between the in-plume and out-of-plume regions offset the effect of volcanic aerosols. For the high-resolution model the impact of the aerosol plume could be assessed directly and always showed an increase in CDNC with a corresponding decrease in Reff when the
volcano was active for the AEROSOL effect. However, the coarse resolution UKESM global model has an unexplained lack of AEROSOL effect response for CDNC and Reff in the third week.

These results highlight that observations need to be treated carefully to assess the impact of meteorology on aerosol-cloud interactions as was done in Peace et al. (2024) using back trajectory analysis. The masking of the aerosol-cloud interactions in the third week, revealed by the modelling, highlights the fact that out of plume meteorological and aerosol conditions
cannot be considered to always be equivalent to in-plume unperturbed conditions. Modelling can play a role in understanding and improving the treatment of these physical processes in weather and climate models.

We also showed that biases in background model fields are important in estimating the quantitative magnitude of aerosol-cloud effects. Increasing the background aerosol amounts closer to reality in these simulations reduced the magnitude of the CDNC and Reff effect. Similarly, the low bias in LWP present in these current simulations may be limiting the impact of
increased CDNC on precipitation evolution and will need to be explored in future work.

Differences in for the magnitude of the effects between the coarser UKESM and finer resolution model suggest that high resolution models are still required for simulating cloud system scale aerosol plume interactions. Simulating multiple volcanic eruption events could provide an avenue for quantitatively testing the ability of models to represent aerosol-cloud interactions in a more thorough manner. It may be possible to use this approach to calibrate the model responses so that they
are more realistic and can then be used to benchmark and improve coarser climate models.

**Code and data availability**

All codes of Python and IDL scripts used in this study *will be uploaded* to Zenodo. Data from simulations and satellite observations *will also be uploaded. (A placeholder for now)*



**Author contributions**

MY, PRF, and DPG designed the study and implemented necessary modifications to the model codes. MY set up the model, performed the simulations, processed and analysed the output data, prepared the figures and tables, and wrote the main part of the manuscript. PRF contributed to the writing, particularly the discussion section, and created one of the tables. MY, PRF, and DPG closely collaborated throughout the project via regular meetings. AHP provided satellite-based observational data and contributed to the relevant parts of the text. JMH and YC offered valuable feedback and advice throughout the

study, from its early stages to final revisions, and suggested improvements to the manuscript draft.

**Competing interests**

We declare no competing interests.

**Acknowledgements**

The symbol of volcano used in figure 2 was generated using PopAi (https://www.popai.pro/). Some of the text has been

refined using ChatGPT (https://chatgpt.com/).

**Financial support**

This research has been supported by the NERC ADVANCE grant (NE/S015671/1).

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
