# Peer review of "Impacts of the Icelandic Holuhraun volcanic eruption on cloud properties using regional model cloud-aerosol simulations"

_EGUsphere, 2025_

## Referee Comment (RC1)

**Review of "*Impacts of the Icelandic Holuhraun volcanic eruption on cloud properties using regional model cloud-aerosol simulations*"**

by Yoshiaoka et al.,

General comments:

In this study, the authors explore the effect of the Holuhraun eruption on cloud properties on a kilometer-scale simulation within the UK Met Office Unified Model. In their setup, the authors employ a detailed cloud microphysical scheme and an interactive aerosol module, making their simulations well-suited to study aerosol-cloud interactions resulting from the Holuhraun eruption. They furthermore nicely outline the importance of considering meteorology and background aerosol conditions in regions that are not directly affected by volcanic aerosol when determining the effect of volcanic aerosols on clouds.
The manuscript is logically structured and well written and merits publication provided that the following comments are addressed.

Specific comments:

- P6, L210-211: Could the authors clarify the rationale behind using a 5 km cloud top threshold? Why was this specific threshold chosen? Given that cloud phase is strongly temperature-dependent, a threshold more directly related to cloud thermodynamics, such as cloud top temperature, may be more appropriate.
- P7, L226-229: The bounding boxes used for the satellite and model data differ in size and shape, which likely introduces biases when comparing in-plume and out-of-plume cloud properties. Since meteorological conditions and, therefore, cloud characteristics vary spatially, using different out-of-plume areas may influence the interpretation of perturbation signals. Furthermore, the authors provide little information on how they define the rectangular boundary that envelops the in-plume regions (I assume it is similar to that in Peace et al., 2024). Could the differences reported in Table 4 stem from different areas of in- and out-of-plume? This is for sure the case for the data from Haghighatnasab et al. (2022), and looking at Peace et al. (2024), the regions are also different.
- P7, L231-232: The threshold of 0.5 DU for defining in-plume regions in the model was chosen to align spatial patterns with satellite data. However, this lower threshold could lead to an underestimation of sulfate burden and, consequently, cloud condensation nuclei (CCN) and cloud droplet number concentration (CDNC) in the model. It would be informative if the authors could show the sensitivity of CDNC to different threshold choices, for example, what changes occur if a 1 DU threshold is applied? Could this partially explain the remaining CDNC underestimation, even after retuning?
- P7, L246-248: The authors claim that their model-derived CDNC is comparable to MODIS-derived CDNC, yet no supporting analysis is provided. MODIS CDNC retrievals are predominantly sensitive to cloud top, while for the model, a liquid-water-weighted vertical average is computed. Although this weighting gives some emphasis to upper layers, it likely results in lower CDNC values compared to satellite retrievals. I know that it is intricate to perform a fully definition-aware comparison between

model and satellite observations (e.g., using a satellite simulator like COSP). I would, nevertheless, ask the authors to somehow show that their CDNC values are comparable to the satellite-derived ones. Alternatively, would it be feasible to extract model CDNC at cloud top for a more definition-consistent comparison?

Minor Remarks:

- P5, L180-187: The authors first perform a 60-day spin-up for the aerosol fields. Are these global simulations nudged to the actual meteorology?
- P7, L244: *"… Section 2.9 …"* This should be 2.8.
- P18, L425: *"… 1.57 …"* It is 1.56 in Fig. 8.
- P21, L476: *"… enhancement for volcano on versus volcano off in the first week in the plume region …"* If I understood correctly, this should be the ERUPTION effect, so I would call it like that.
- P21, L482: *"… the in-plume and out-of-plume regions …"* If I understood correctly, this should be the LOCATION effect.

References:

Haghighatnasab, M., Kretzschmar, J., Block, K., and Quaas, J.: Impact of Holuhraun volcano aerosols on clouds in cloud-system-resolving simulations, Atmos. Chem. Phys., 22, 8457–8472, https://doi.org/10.5194/acp-22-8457-2022, 2022.

Peace, A. H., Chen, Y., Jordan, G., Partridge, D. G., Malavelle, F., Duncan, E., and Haywood, J. M.: In-plume and out-of-plume analysis of aerosol–cloud interactions derived from the 2014–2015 Holuhraun volcanic eruption, Atmos. Chem. Phys., 24, 9533–9553, https://doi.org/10.5194/acp-24-9533-2024, 2024.

---

## Referee Comment (RC2)

Review

This paper reports on aerosol-cloud interactions observed from satellite measurements and simulated with a high-resolution regional model (Unified Model) using the Holuhraun eruption as an opportunistic experiment. The authors find strong aerosol and "total" effects on cloud droplet number concentrations and effective radius due to the volcanic eruption with minimal and statistically insignificant responses to cloud fraction and liquid water path. The high-resolution of the UM allowed for sufficient realization of the cloud responses to the eruption as compared to more coarse model comparisons that were not as successful. However, uncertainties in the simulation of background aerosols in the UM may complicate these results. This is a fairly straightforward and clearly written paper that carefully walks through the differences in observed and simulated effects and the limitations of each data source in evaluating these differences. I believe this paper is suitable for this journal as it uses innovative approaches for quantifying aerosol-cloud interactions from an opportunistic experiment and provides recommendations for improving future efforts. The authors should consider the following minor comments, questions, and recommendations to improve the work before it should be published.

- Are the authors able to add titles to the sets of columns in Figure 1? Left: Simulated, Right: Satellite? This would allow for a more accessible direct comparison between the plots.

- The amount of underprediction in CDNC in the model seems rather notable. Is the magnitude of the CDNC underprediction similar to previous work using this and other models? Does the claim that background aerosols are likely to blame for this discrepancy consistent with underpredictions in other work or a known issue with this model? How does the out-of-plume model AOD compare to the satellite AOD to support this claim?

- In the discussion and conclusions, can the authors posit on the potential meteorological covariabilities that my lead to a reduction in the TOTAL and LOCATION effects for CDNC and Reff in week 3? Why did the authors not consider these effects and the other mentioned effects using cloud-controlling factors for this purpose?

- Can the authors briefly speak to some (if any) of the microphysical parameterization scheme differences that could lead to differences in effects between model datasets?

- Lines 468-469: has evidence of this semi-direct effect been suggested or shown in similar previous work? If so, the authors should provide citation here. If not, I still feel it appropriate for the authors to provide some citation to support this point of discussion.

- Lines 473-474 (answer to intro question 1): can the authors please provide a quantification of the CDNC and Reff increases/reductions?

---

## Author Comment (AC1)

**Responses to referee 1**

**General comments:**

In this study, the authors explore the effect of the Holuhraun eruption on cloud properties on a kilometer-scale simulation within the UK Met Office Unified Model. In their setup, the authors employ a detailed cloud microphysical scheme and an interactive aerosol module, making their simulations well-suited to study aerosol-cloud interactions resulting from the Holuhraun eruption. They furthermore nicely outline the importance of considering meteorology and background aerosol conditions in regions that are not directly affected by volcanic aerosol when determining the effect of volcanic aerosols on clouds.

The manuscript is logically structured and well written and merits publication provided that the following comments are addressed.

**Specific comments:**

• P6, L210-211: Could the authors clarify the rationale behind using a 5 km cloud top threshold? Why was this specific threshold chosen? Given that cloud phase is strongly temperature-dependent, a threshold more directly related to cloud thermodynamics, such as cloud top temperature, may be more appropriate.

We thank the reviewer for this comment. The cloud top threshold of 5 km was chosen as a practical criterion to separate lower tropospheric liquid clouds from upper tropospheric ice clouds such as cirrus. A domain mean height profile of liquid water content shows that more than 99% of the liquid water resides below 4 km (see the plot below). Therefore, a 5 km threshold effectively captures the relevant population of marine liquid clouds, and we do not expect the precise choice of this upper limit to strongly affect our results. Furthermore, the same threshold was applied consistently to both satellite observations and model simulations, so any potential bias introduced by this choice is mitigated in the comparisons. We added the following sentence in the main text;

This 5 km threshold was chosen to separate lower-tropospheric liquid clouds from upper-tropospheric ice clouds such as cirrus. In our simulations, more than 99% of liquid water resides below 4 km, so varying this threshold within a reasonable range (e.g. 4–6 km) would not substantially affect the results.

• P7, L226-229: The bounding boxes used for the satellite and model data differ in size and shape, which likely introduces biases when comparing in-plume and out-of plume cloud properties. Since meteorological conditions and, therefore, cloud characteristics vary spatially, using different out-of-plume areas may influence the interpretation of perturbation signals. Furthermore, the authors provide little information on how they define the rectangular boundary that envelops the in-plume regions (I assume it is similar to that in Peace et al., 2024). Could the differences reported in Table 4 stem from different areas of in- and out-of-plume? This is for sure the case for the data from Haghighatnasab et al. (2022), and looking at Peace et al. (2024), the regions are also different.

The bounding boxes are defined using satellite-retrieved and model-simulated column  $SO_2$  burdens, which differ from each other. These reflect the differences in atmospheric transport and dispersion between the real atmosphere and the model. Therefore, the boxes are internally consistent within each dataset and are suitable for comparing the effects of volcanic plume in a self-consistent manner. Using identical geographic regions for model and observations would lead to include plume effects in the out-of-plume regions and vice versa.

We have expanded the description of choice of regions as follows:

"In-plume" and "out-of-plume" regions were determined by selecting the minimum and maximum east-west and north-south extents of the plume, following the methodology of Peace et al. (2024).

• P7, L231-232: The threshold of 0.5 DU for defining in-plume regions in the model was chosen to align spatial patterns with satellite data. However, this lower threshold could lead to an underestimation of sulfate burden and, consequently, cloud condensation nuclei (CCN) and cloud droplet number concentration (CDNC) in the model. It would be

informative if the authors could show the sensitivity of CDNC to different threshold choices, for example, what changes occur if a 1 DU threshold is applied? Could this partially explain the remaining CDNC underestimation, even after retuning?

We share the referee's concern about the difference in thresholds applied to satellite observations and model simulations, which could affect the analyses. To address this, we generated plots equivalent to Fig. 4 using a 1 DU threshold for the simulation data instead of 0.5 DU.

These plots show slightly higher CDNC inside the plume, but we confirm that this difference does not significantly affect our analyses or conclusions.

We added these plots in Fig. S2 and added this sentence in the main text. "While using a 0.5 DU threshold provides better spatial agreement between the satellite and model aerosol fields, using a 1.0 DU threshold does not qualitatively or quantitatively (up to 18% increase in Volc.In CDNC) change the results (see figure S2)."

• P7, L246-248: The authors claim that their model-derived CDNC is comparable to MODIS-derived CDNC, yet no supporting analysis is provided. MODIS CDNC retrievals are predominantly sensitive to cloud top, while for the model, a liquid-water weighted vertical average is computed. Although this weighting gives some emphasis to upper layers, it likely results in lower CDNC values compared to satellite retrievals. I know that it is intricate to perform a fully definition-aware comparison between model and satellite observations (e.g., using a satellite simulator like COSP). I would, nevertheless, ask the authors to somehow show that their CDNC values are comparable to the

satellite-derived ones. Alternatively, would it be feasible to extract model CDNC at cloud top for a more definition-consistent comparison?

To demonstrate that using liquid water content weighted CDNC to compare to MODIS is acceptable we show some analysis where we have selected columns where condensate in any grid box below 2 km greater than 0.5 g m-3 with clear sky above. This picks out columns that are predominantly stratus cloud and eliminates the deeper frontal clouds that will present ice to MODIS. Composite plots of CDNC and LWC in the volcano plume area with volcano on and off show that the LWC weighting will emphasise where the peak grid mean CDNC values are for comparison with MODIS.

The MODIS retrieval of CDNC assumes a constant CDNC throughout the depth of the cloud, rather than retrieving cloud top CDNC per se. However, the CDNC retrieval is very sensitive to the effective radius retrieval, which is weighted towards cloud top (in terms of optical depth from cloud top). It also assumes a linear increase in liquid water content (LWC) with height, so that LWC is highest at cloud top under this assumption.

Our approach of weighting the model CDNC by LWC is a compromise to ensure that the vertical levels with high LWC, which are most important for the cloud optical depth and radiative effect, contribute most to the mean CDNC value. It also reduces the influence of spurious thin (low LWC) cloud levels.

If the model clouds behave similarly to the assumptions made for the MODIS CDNC retrieval, then the exact vertical weighting would not matter much, since CDNC would be vertically constant. If, however, the vertical profiles of model clouds (or real clouds) are different from these assumptions, using only model cloud-top CDNC or trying to replicate the retrieval's vertical weighting may not be the most suitable approach. Further work would be needed to investigate this, but such an analysis would be complex and beyond the scope of this study.

**Minor Remarks:**

• P5, L180-187: The authors first perform a 60-day spin-up for the aerosol fields. Are these global simulations nudged to the actual meteorology?

The global meteorology is initialised every 24 hours from Met Office analyses data. We expect that this has a similar effect to nudging although the relaxation time is usually

shorter in nudging (typically 6 hours). We initially used 30 day spin-up period but found a drift in the simulation and hence made it longer. By the end of 60 day spin-up, the simulation appeared to have reached steady state as shown below (sulphate mass mixing ratio (left) and particle number mixing ratio (right) in Aitken mode in 30-60N).

• P7, L244: "... Sec'on 2.9 ..." This should be 2.8.

We thank the referee for pointing out the error. It has been corrected.

• P18, L425: "... 1.57 ..." It is 1.56 in Fig. 8.

We appreciate the referee's careful reading and have corrected this error.

• P21, L476: "... enhancement for volcano on versus volcano off in the first week in the plume region ..." If I understood correctly, this should be the ERUPTION effect, so I would call it like that.

Thank you for bringing this to our attention. We have added '(ERUPTION effect)' in the text. Likewise, "out of plume to in-plume values" should be the TOTAL effect. Also the ERUPTION effect in the first week should actually be ~20% enhancement instead of ~10%. So the text has become like this;

"The UM liquid water path remains largely unchanged (except the third week) when comparing out of plume to in-plume values (TOTAL effect) but does show a ~20% enhancement for volcano on versus volcano off (ERUPTION effect) in the first week in the plume region."

• P21, L482: "... the in-plume and out-of-plume regions ..." If I understood correctly, this should be the LOCATION effect.

The referee is correct in that the comparison between in-plume and out-of-plume regions will give us the LOCATION effect. However, this part of the text talks about the

differences in the meteorological environments in two regions. We consider it as the cause of the LOCATION effect and so would not call it the LOCATION effect itself.

**References**

Haghighatnasab, M., Kretzschmar, J., Block, K., and Quaas, J.: Impact of Holuhraun volcano aerosols on clouds in cloud-system-resolving simulations, Atmos. Chem. Phys., 22, 8457–8472, https://doi.org/10.5194/acp-22-8457-2022, 2022.

Peace, A. H., Chen, Y., Jordan, G., Partridge, D. G., Malavelle, F., Duncan, E., and Haywood, J. M.: In-plume and out-of-plume analysis of aerosol–cloud interactions derived from the 2014–2015 Holuhraun volcanic eruption, Atmos. Chem. Phys., 24, 9533–9553, https://doi.org/10.5194/acp-24-9533-2024, 2024.

---

## Author Comment (AC2)

**Responses to referee 2**

**Review**

This paper reports on aerosol-cloud interactions observed from satellite measurements and simulated with a high-resolution regional model (Unified Model) using the Holuhraun eruption as an opportunistic experiment. The authors find strong aerosol and "total" effects on cloud droplet number concentrations and effective radius due to the volcanic eruption with minimal and statistically insignificant responses to cloud fraction and liquid water path. The high-resolution of the UM allowed for sufficient realization of the cloud responses to the eruption as compared to more coarse model comparisons that were not as successful. However, uncertainties in the simulation of background aerosols in the UM may complicate these results. This is a fairly straightforward and clearly written paper that carefully walks through the differences in observed and simulated effects and the limitations of each data source in evaluating these differences. I believe this paper is suitable for this journal as it uses innovative approaches for quantifying aerosol-cloud interactions from an opportunistic experiment and provides recommendations for improving future efforts. The authors should consider the following minor comments, questions, and recommendations to improve the work before it should be published.

• Are the authors able to add titles to the sets of columns in Figure 1? Left: Simulated, Right: Satellite? This would allow for a more accessible direct comparison between the plots.

We thank the referee's suggestion. We added 'Simulated' and 'Satellite' at the top of the columns in Fig 1.

• The amount of underprediction in CDNC in the model seems rather notable. Is the magnitude of the CDNC underprediction similar to previous work using this and other models? Does the claim that background aerosols are likely to blame for this discrepancy consistent with underpredictions in other work or a known issue with this model? How does the out-of-plume model AOD compare to the satellite AOD to support this claim?

We agree that the aerosol concentrations are underestimated in our simulations. An underestimation of CDNC over northern North Atlantic has been reported in Grosvenor and Carslaw (2020) for the UKESM global model simulation. Our recent analysis with newer version of model (UKESM1.1) indicates a low bias of N50 by about 40% (NMBF~-75%) in North Atlantic (see the blue bar (BLN) in the figure below). Even stronger low bias in particle number concentration has been seen in UM\_UKCA regional simulations off the coast of Portugal (Yoshioka et al., personal communication).

The low bias in our original simulation can also be seen in AOD, although we note that AOD may not be a good predictor of CDNC because it is dominated by second moment of the size distribution ( $\int n^*r^2$ ). Table below shows area-averaged AOD in the simulations used in this study compared to MODIS AOD, both at 550 nm, together with relative (%) differences (simulation/MODIS \* 100) for the first 6 days. The selected region (50-62N; 40-15W) covers the northern North Atlantic SW of Iceland, mostly outside the volcanic plume (containing regions out of plume and out of bounds). This indicates that the original simulation has a low bias except for the first two days, and that this bias has been somewhat reduced in the enhanced simulation.

However, we note that increasing the background aerosol concentration by changing model settings and perturbing the model parameters did not affect the results of comparing volcano on and off simulations.

Nevertheless, we are fully aware of the pressing need to improve the background aerosol and CDNC in the UM-UKCA model. We added the following sentences in the second paragraph of section 3.1:

General underestimation is considered likely due to biases in background aerosols or an underestimate in the treatment of the activation of cloud droplets than a bias in the Holuhraun volcanic emission implemented in the model. The use of alternate activation schemes is a subject of ongoing investigation.

| date     | original | %diff | enhanced | %diff | MODIS |
|----------|----------|-------|----------|-------|-------|
| 20140901 | 0.148    | 21%   | 0.152    | 24%   | 0.122 |
| 20140902 | 0.096    | -4%   | 0.102    | 2%    | 0.100 |
| 20140903 | 0.087    | -23%  | 0.098    | -14%  | 0.113 |
| 20140904 | 0.077    | -25%  | 0.084    | -18%  | 0.103 |
| 20140905 | 0.088    | -26%  | 0.092    | -23%  | 0.119 |
| 20140906 | 0.087    | -46%  | 0.089    | -44%  | 0.159 |

• In the discussion and conclusions, can the authors posit on the potential meteorological covariabilities that my lead to a reduction in the TOTAL and LOCATION effects for CDNC and Reff in week 3? Why did the authors not consider these effects and the other mentioned effects using cloud-controlling factors for this purpose?

We thank the referee for raising this important point. We calculated the lower tropospheric stability (LTS) as the difference between the potential temperatures at the 700 hPa level and the surface. The figure below shows box-and-whisker plots of LTS within and outside the plume regions for both the Volc and NoVolc simulations over four weeks.

The results indicate that LTS was higher inside the plume than outside in week 3, in contrast to the other weeks. They also show that this pattern is insensitive to whether the volcano is on or off in the simulations. Therefore, mesoscale variations in stability are likely to have contributed to the masking effect observed in week 3. We added these plots in figure 9 and the following sentences in the fourth paragraph of section 4:

Examination of the lower tropospheric stability (LTS), for the high resolution UM, in and out of the plume and for volcano on and off is provided in figure 9. This indicates that LTS is greater inside the plume than outside in week 3, in contrast to the other weeks, and that this result is insensitive to whether the volcano is on or off in the simulations. Therefore, the mesoscale stability variations are likely to have been the source of the masking effect seen in week 3.

• Can the authors briefly speak to some (if any) of the microphysical parameterization scheme differences that could lead to differences in effects between model datasets?

Section 2 describes the CASIM cloud microphysics scheme used in this study. The UKESM simulation analyzed by Peace et al. (2024) uses a single-moment bulk scheme (Wilson and Ballard, 1999; Sellar et al., 2019) in which droplet number concentration is diagnosed rather than prognosed. In contrast, CASIM explicitly predicts droplet number and mass, allowing a more direct representation of aerosol–cloud interactions (Grosvenor et al., 2017; Field et al., 2023). This difference may contribute, in addition to the resolution difference discussed in Section 4, to the weaker CDNC and Reff responses in UKESM compared to our simulations.

ICON also uses a two-moment bulk scheme (Seifert and Beheng, 2006) similar to CASIM. However, since their results cover only the first week and there are also differences in the atmospheric models and experimental designs used, it is difficult to attribute the simulated differences in cloud properties to differences in the cloud microphysics schemes.

• Lines 468-469: has evidence of this semi-direct effect been suggested or shown in similar previous work? If so, the authors should provide citation here. If not, I still feel it appropriate for the authors to provide some citation to support this point of discussion.

We understand the reviewer's concern about the impact of volcanic aerosols on the cloud field outside the plume. There is indeed precedent in the literature for semi-direct or circulation-mediated effects from aerosol plumes. For example, Diamond et al. (2022) reported that smoke over the southeastern Atlantic not only caused local semi-direct effects but also altered the large-scale atmospheric thermal structure, thereby affecting cloud properties over a broader region.

We have added the following sentence: "This is similar to the effect of smoke over the southeastern Atlantic altering the large-scale atmospheric thermal structure and thereby cloud properties, as reported by Diamond et al. (2022)."

• Lines 473-474 (answer to intro question 1): can the authors please provide a quantification of the CDNC and Reff increases/reductions?

We modified the text and included the ranges of changes in CDNC and Reff as follows; "The modelling results show an increase in droplet number by a factor of 1.6 to 2.6 and a reduction in effective radius by 1.6 to 2.7  $\mu$ m, except during the third week. These findings are consistent with the direction of changes seen in satellite observations and previous modelling studies, though the exact magnitudes vary among datasets."

**References**

Diamond, M. S., Saide, P. E., Zuidema, P., Ackerman, A. S., Doherty, S. J., Fridlind, A. M., Gordon, H., Howes, C., Kazil, J., Yamaguchi, T., Zhang, J., Feingold, G., and Wood, R.: Cloud adjustments from large-scale smoke–circulation interactions strongly modulate the southeastern Atlantic stratocumulus-to-cumulus transition, Atmos. Chem. Phys., 22, 12113–12151, https://doi.org/10.5194/acp-22-12113-2022, 2022.

Field, P.R., Hill, A., Shipway, B., Furtado, K., Wilkinson, J., Miltenberger, A., et al. (2023) Implementation of a double moment cloud microphysics scheme in the UK met office regional numerical weather prediction model. Quarterly Journal of the Royal Meteorological Society, 149(752), 703–739. Available from: https://doi.org/10.1002/qj.4414

Grosvenor, D. P. and Carslaw, K. S.: The decomposition of cloud–aerosol forcing in the UK Earth System Model (UKESM1), Atmos. Chem. Phys., 20, 15681–15724, https://doi.org/10.5194/acp-20-15681-2020, 2020.

Grosvenor, D. P., Field, P. R., Hill, A. A., and Shipway, B. J.: The relative importance of macrophysical and cloud albedo changes for aerosol-induced radiative effects in closed-cell stratocumulus: insight from the modelling of a case study, Atmos. Chem. Phys., 17, 5155–5183, https://doi.org/10.5194/acp-17-5155-2017, 2017. a

Sellar, A. A., Jones, C. G., Mulcahy, J. P., Tang, Y., Yool, A., Wiltshire, A., et al. (2019). UKESM1: Description and evaluation of the U.K. Earth System Model. *Journal of Advances in Modeling Earth Systems*, *11*, https://doi.org/10.1029/ 2019MS001739 Seifert, A., Beheng, K.: A two-moment cloud microphysics parameterization for mixed-phase clouds. Part 1: Model description. Meteorol. Atmos. Phys. 92, 45–66. https://doi.org/10.1007/s00703-005-0112-4, 2006.

Wilson, D.R. and Ballard, S.P.: A microphysically based precipitation scheme for the UK meteorological office unified model. Q.J.R. Meteorol. Soc., 125: 1607-1636. https://doi.org/10.1002/qj.49712555707, 1999.